# Multidimensional analyses reveal modulation of adaptive and innate immune subsets by tuberculosis vaccines

Virginie Rozot [1,11✉], Elisa Nemes [1,11], Hennie Geldenhuys[1], Munyaradzi Musvosvi[1], Asma Toefy[1], Frances Rantangee[1], Lebohang Makhethe[1], Mzwandile Erasmus[1], Nicole Bilek[1], Simbarashe Mabwe[1], Greg Finak [2], William Fulp[2], Ann M. Ginsberg [3], David A. Hokey[3], Muki Shey [4], Sanjay Gurunathan[5], Carlos DiazGranados[5], Linda-Gail Bekker[6], Mark Hatherill[1], Thomas J. Scriba [1✉] & The C-040-404 Study Team*

We characterize the breadth, function and phenotype of innate and adaptive cellular responses in a prevention of *Mycobacterium tuberculosis* infection trial. Responses are measured by whole blood intracellular cytokine staining at baseline and 70 days after vaccination with H4:IC31 (subunit vaccine containing Ag85B and TB10.4), Bacille Calmette-Guerin (BCG, a live attenuated vaccine) or placebo (n = ~30 per group). H4:IC31 vaccination induces Ag85B and TB10.4-specific CD4 T cells, and an unexpected NKT_like subset, that expresses IFN-γ, TNF and/or IL-2. BCG revaccination increases frequencies of CD4 T cell subsets that either express Th1 cytokines or IL-22, and modestly increases IFNγ-producing NK cells. In vitro BCG re-stimulation also triggers responses by donor-unrestricted T cells, which may contribute to host responses against mycobacteria. BCG, which demonstrated efficacy against sustained *Mycobacterium tuberculosis* infection, modulates multiple immune cell subsets, in particular conventional Th1 and Th22 cells, which should be investigated in discovery studies of correlates of protection.

[1] South African Tuberculosis Vaccine Initiative, Institute of Infectious Disease & Molecular Medicine and Division of Immunology, Department of Pathology, University of Cape Town, Cape Town, South Africa. [2] Fred Hutchinson Cancer Research Center (FHCRC), Seattle, WA, USA. [3] AERAS, Rockville, MD, USA. [4] Aeras South Africa Endpoint Assay Laboratory, Cape Town, South Africa. [5] Sanofi Pasteur, Swiftwater, PA, USA. [6] The Desmond Tutu HIV Centre, University of Cape Town, Cape Town, South Africa. [11]These authors contributed equally: Virginie Rozot, Elisa Nemes. [12]Deceased: Robert Ryall. *A list of authors and their affiliations appear at the end of the paper. ✉email: virginie.rozot@uct.ac.za; thomas.scriba@uct.ac.za

A vaccine that can prevent infection with *Mycobacterium tuberculosis* (Mtb) could have a major impact on the tuberculosis (TB) epidemic[1].

We recently completed a phase IIb randomized, controlled, partially blinded trial (C-040-404), which aimed to determine the safety, immunogenicity, and efficacy of H4:IC31 vaccination or Bacille Calmette-Guerin (BCG) revaccination to prevent QuantiFERON TB Gold In-Tube (QFT) conversion in previously BCG vaccinated, QFT-negative adolescents[2].

The H4:IC31 vaccine candidate comprises a fusion protein of mycobacterial antigens Ag85B and TB10.4 formulated in the IC31 adjuvant. H4:IC31 vaccination was protective in animal models[3–5] and administration in humans has been well tolerated with acceptable safety profiles and typically induces predominantly polyfunctional CD4 Th1 cell responses[6,7].

By contrast, the live attenuated, whole-cell BCG vaccine comprises a complex array of lipid, polysaccharide, and protein antigens and has been shown to induce potent MHC-restricted Th1-cytokine-expressing CD4 T cell responses as well as donor-unrestricted T cell (DURT) and innate cell responses[8–11].

Neither vaccine protected against the primary C-040-404 trial endpoint, initial infection with Mtb, defined as conversion to a positive QFT at the manufacturers' threshold ($\geq$0.35 IU/mL). Efficacy of H4:IC31 against sustained QFT conversion, defined as QFT conversion without reversion to a negative QFT for three consecutive tests, was 30.5% (95% CI −15.8 to 59.3, $p = 0.16$), and efficacy against conversion at the exploratory cut-off of 4 IU/mL was 34.5% (95% CI −12.1 to 62.3, $p = 0.13$). Neither of these efficacy estimates met the standard 95% CI criteria for statistical significance. By contrast, BCG revaccination demonstrated significant protection against sustained QFT conversion with vaccine efficacy of 45.4% (95% CI 6.4–68.1, $p = 0.03$), as well as protection against initial QFT conversion at a threshold of 4 IU/mL of 45.1% (95% CI 3.8–69.3, $p = 0.04$)[2].

To inform which immune response features may contribute to these observed vaccine effects, we sought to comprehensively characterize the breadth, function, and phenotype of innate and adaptive cellular immune responses boosted by H4:IC31 vaccination or BCG revaccination in the C-040-404 trial.

The immunological mechanisms that govern control of Mtb infection in humans are incompletely understood. Numerous studies have highlighted a fundamental role of CD4 T cells in protective immunity against Mtb. Experiments in animal models of TB have shown that depletion of CD4 T cells leads to uncontrolled bacterial growth and clinical deterioration[12,13]. In addition, HIV-infected individuals with CD4 T cell depletion are at significantly increased risk of TB disease, possibly because HIV has been shown to preferentially impair Mtb-specific T cells[14–16]. Intact Mtb-specific Th1 responses are necessary for successful control of mycobacterial infection, since patients with inborn mutations in genes of the Th1 pathway, such as STAT1, interleukin-12 (IL-12), IL-12R, interferon-$\gamma$ (IFN$\gamma$), or IFN$\gamma$R, are highly susceptible to mycobacterial disease due to Mtb, non-tuberculous mycobacteria, or BCG[17]. Th1 CD4 T cells produce cytokines such as IFN$\gamma$ or tumor necrosis factor (TNF), which contribute to recruitment and activation of macrophage antimicrobial activity at the site of infection[18]. It follows that many TB vaccine strategies aim to induce polyfunctional and long-lasting CD4 Th1 cell responses[19,20].

Four studies in the non-human primate model of TB have pointed to antigen-specific CD4 (and possibly CD8) T cells that express Th1/Th17 cytokines in bronchoalveolar lavage or lung tissue as candidate correlates of protection against TB[21–24]. The first novel TB vaccine candidate to reach efficacy testing in human infants since BCG, MVA85A, boosted Ag85A-specific, polyfunctional and durable memory Th1 and low-level Th17

responses in the peripheral blood that were primed by BCG[25,26]. However, MVA85A vaccination provided no protection against TB disease above that afforded by newborn BCG vaccination[27]. These data, along with other studies[28], suggest that antigen-specific Th1 responses are necessary but not sufficient for the control of Mtb (reviewed in ref. [29]).

The primary immunogenicity outcome of the C-040-404 trial, namely frequencies of CD4 T cells expressing IFN$\gamma$ and/or IL-2, was reported previously[2]. Here we sought to characterize the immune responses induced by H4:IC31 and BCG more comprehensively to explore the breadth of responding immune subsets and their functional and phenotypic attributes. We proposed that identification of novel, or confirmation of known, responding cell subsets and their attributes will inform approaches and hypotheses for follow-up studies to evaluate correlates of protection. We hypothesized that the variety of immune responses induced by a subunit (H4:IC31) or a whole-cell (BCG) TB vaccine would be different, where H4:IC31-specific responses would be predominated by Th1-cytokine-expressing CD4 T cells, while BCG would induce a broader range of immune subsets and functions.

## Results

**Vaccine-specific CD4 Th1 cell responses detected in peripheral blood mononuclear cells (PBMCs) and whole blood (WB) are correlated.** Vaccine immunogenicity was measured by PBMCs intracellular cytokine staining (ICS)[2] and WB-ICS assays. Frequencies of antigen-specific Th1 cells (i.e. CD4+ T cells producing any combination of IFN$\gamma$, IL-2, and/or TNF) measured by PBMC and WB-ICS assays correlated significantly ($r = 0.59$, $p < 0,0001$, Supplementary Fig. 1a). Of note, Th22 responses did not correlate between the two assays and were lower and virtually undetectable in PBMCs (Supplementary Fig. 1b, c). We also observed that post-vaccination Th17 responses were virtually undetectable after restimulation with Ag85B or TB10.4 in H4:IC31 arm participants or BCG restimulation in BCG arm participants (Supplementary Fig. 1D). The use of different antibody clones for the PBMC and WB-ICS assays (Supplementary Table 1) may partly explain these observations.

Regardless, the WB-ICS assay was more suitable to detect responses by a broad range of leukocytes to complex antigens, such as BCG[30], and thus we report on data generated by WB-ICS assay hereafter.

**Data analysis strategy to evaluate vaccine-induced responses.** Immunogenicity analyses performed in clinical vaccine trials generally apply targeted and hypothesis-driven approaches in order to quantify and compare subsets of cells expected to be induced by vaccination[31–35]. In the TB field, most immunogenicity analyses performed so far have targeted CD4 T cell and antibody responses, while it is widely recognized that natural immunity to Mtb might encompass a broader set of immune responses[29]. We designed a flow cytometry gating strategy (Supplementary Fig. 2) to identify all lymphocytes that respond to stimulation with either Ag85B and TB10.4 or BCG by producing cytokines in whole blood. t-Stochastic neighbor embedding (tSNE)[36], a nonlinear dimensionality reduction visualization tool, was used to visualize the relative contribution of each cell subset to the overall functional (i.e. cytokine expressing) response to stimulation. CITRUS[37] was used to identify clusters of cytokine-expressing cells that were significantly induced by BCG revaccination [by comparing immune responses detected before (day 0) and after (day 70) BCG revaccination] and that were different between placebo and BCG recipients at day 70. CITRUS cluster identity was validated by manual gating. Functional profiles of

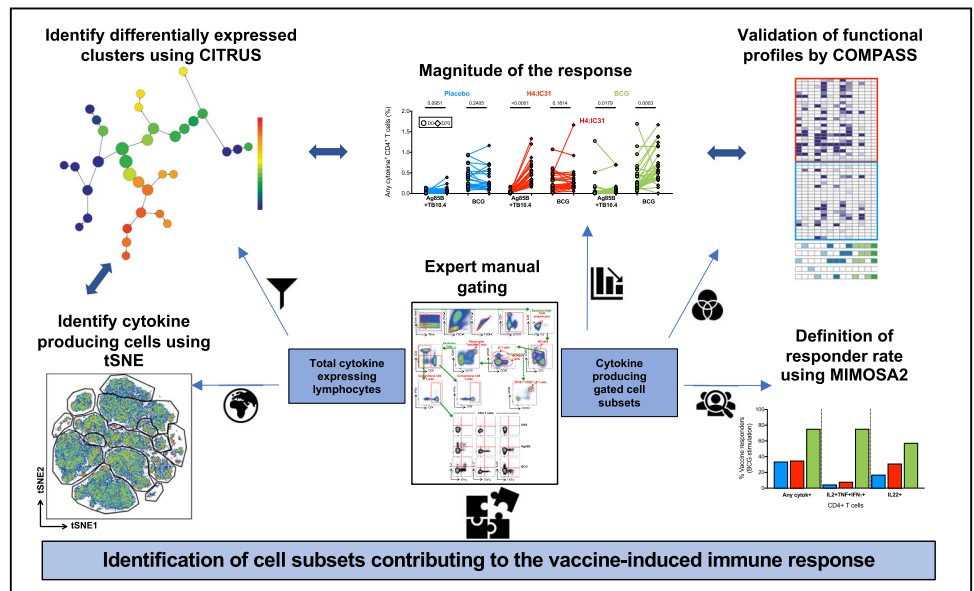

**Fig. 1 Schematic representation of the analysis pipeline and approaches used to characterize the immune response after in vitro stimulation with Ag85B and TB10.4 or BCG in trial participants.** Lymphocytes expressing any cytokines, selected by conventional flow cytometry gating in stimulated and unstimulated samples, were subjected to unsupervised analyses (left) using tSNE and CITRUS, or to targeted analyses (right) where subsets were manually gated and cytokine-co-expression subsets were analyzed by comparing their frequencies directly, or by using COMPASS (functionality and polyfunctionality assessments) or MIMOSA2 (to assign vaccine responder and non-responder status). Double sided arrows show the analyses used in parallel to validate observations with multiple strategies. Single arrows indicate the flow of analysis.

these cell subsets were further assessed using COMPASS[38] and vaccine responsiveness over baseline was evaluated by a novel algorithm, mixture models for single-cell assays 2 (MIMOSA2) (see Fig. 1 and "Methods").

**H4:IC31 induced Th1 and NKT_{like} cell responses.** Previous studies have demonstrated induction of polyfunctional Th1-cytokine-expressing CD4 T cells by H4:IC31 vaccination[6,7]. However, whether other subsets of innate or unconventional T cells are modulated directly or indirectly, for example, through cytokine-mediated by-stander activation by H4:IC31 has not been investigated. We applied tSNE[36,39] to identify all cell subsets that expressed cytokines in response to stimulation with Ag85B and TB10.4 peptide pools in H4:IC31 recipients (Fig. 2a, b and Supplementary Fig. 3). Responses measured after Ag85B and TB10.4 stimulation were similar in terms of magnitude and cell subset composition. CD4 T cells represented 43.6% (95% CI 35.4–51.8) of the total Ag85B-specific and 26.6% (95% CI 19.6–33.5) of the TB10.4-specific cytokine-expressing cells (Fig. 2a, b and Supplementary Fig. 5a). Even though these CD4 T cells did not sub-cluster in the tSNE plot by their CD26 and CD161 expression, known to be associated with T cell activation[40,41], a large proportion of CD4 T cells expressed either CD26 or CD161 (Supplementary Fig. 3a, b).

Another prominent cytokine-producing cell population observed after Ag85B and TB10.4 stimulation (35.8%, 95% CI 28.4–43.3 and 46.1%, 95% CI 38–54.2, respectively) was a CD56_{int} NK cell subset that mostly expressed IFNγ (Fig. 2a, b and Supplementary Fig. 3). This IFNγ-expressing NK cell population appeared to be non-specifically activated in vivo, as it was also observed at day 0 and was detected in the unstimulated control, suggesting constitutive cytokine expression or a technical artifact. A small CD56_{hi} NK cell subset (1.8%, 95% CI 0.5–3.2 and 3.1%, 95% CI 0.9–5.4 of the total response in response to Ag85B and TB10.4, respectively) was also identified. Since a significant increase at day 70 was not observed, this subset did not appear to be modulated by H4:IC31 vaccination (Supplementary Fig. 5b).

NKT_{like} cells, identified as CD3+ and CD56+, represented 6.4% (95% CI 3.6–9.3) and 7.1% (95% CI 4.1–10.1) of the total response to Ag85B and TB10.4, respectively (Fig. 2a, b and Supplementary Figs. 3 and 5c), expressed mostly IFNγ and TNF and were significantly increased after H4:IC31 vaccination (Supplementary Fig. 4a). COMPASS analysis comparing day 70 over day 0 immune profiles confirmed induction of an NKT_{like} subset expressing IFNγ, TNF, and IL-2 after H4:IC31 vaccination in 20 out of 26 participants (Supplementary Fig. 4b).

Of note, γδ T cells represented 2.2% (95% CI 0–4.5) and 2.4% (95% CI 0–4.9) of the total Ag85B- and TB10.4-responding cells, respectively, and phenotypically defined MAIT cells 0.6% (95% CI 0.2–1) and 1.1% (95% CI 0.3–1.9, Fig. 2a, b and Supplementary Fig. 5a, b). Frequencies of both subsets increased significantly after H4:IC31 vaccination (Supplementary Fig. 4a), but COMPASS analysis showed that these increases were only seen in very few participants (Supplementary Fig. 4b).

Finally, tSNE also revealed three small cytokine-producing cell populations that our panel of markers could not identify (Fig. 2a, b and Supplementary Fig. 3).

Altogether, these data show vaccine-mediated induction of expected H4:IC31-specific CD4 T cell responses, but also pointed out induction of NKT_{like} cells producing various combinations of Th1 cytokines.

Due to the low frequencies of cytokine-producing cells in response to Ag85B and TB10.4 before vaccination and in placebo recipients, we were not able to use an unsupervised method to identify differential abundance of cytokine-expressing subsets compared to H4:IC31 post-vaccination responses. Regardless, we further analyzed conventional CD4 T cells by manual gating. Frequencies of Ag85B and TB10.4-specific CD4 T cells were low at day 0 and significantly boosted by H4:IC31 vaccination (Fig. 2c). These H4:IC31-induced responses were almost exclusively comprised of Th1-cytokine-expressing CD4 T cells; IL-17- and IL-22-expressing CD4 T cells were very low or undetectable (Fig. 2d), although frequencies of IL-17-expressing

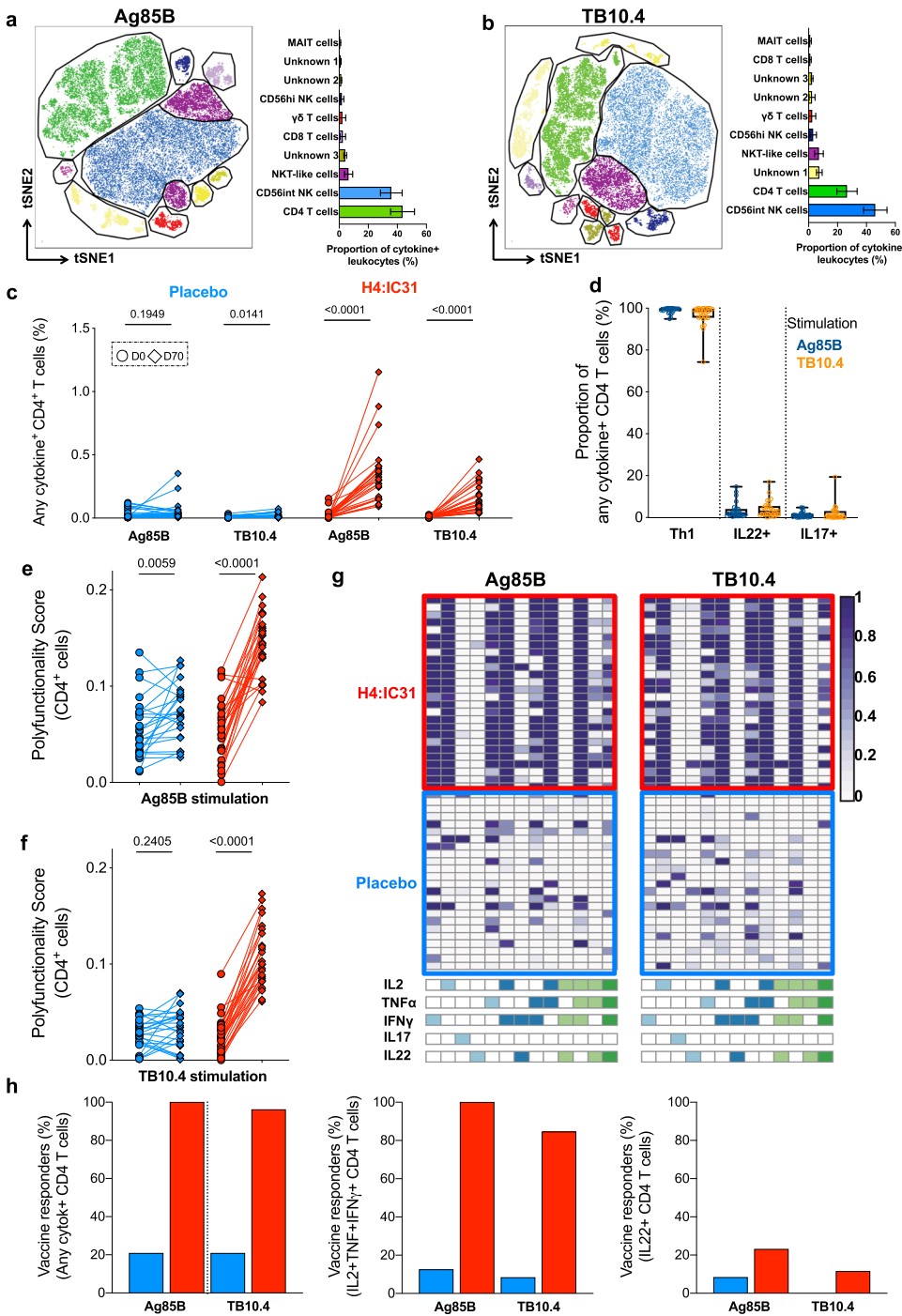

4    COMMUNICATIONS BIOLOGY | (2020)3:563 | https://doi.org/10.1038/s42003-020-01288-3 | www.nature.com/commsbio

CD4 T cells were significantly boosted by H4:IC31 vaccination (Supplementary Fig. 4c). Cytokine-co-expression profiles in response to Ag85B and TB10.4 were not different (Fig. 2d and Supplementary Fig. 4d).

Next, we assessed vaccine-induced changes in cytokine co-expression and polyfunctionality of antigen-specific CD4 T cell by COMPASS[38]. Relative to the pre-vaccination response, Ag85A and TB10.4-specific CD4 T cell polyfunctionality increased markedly after vaccination in H4:IC31 vaccinees (Fig. 2e, f). To further dissect these functional profiles, we also computed posterior probabilities of differences in cytokine co-expression by CD4 T cells at day 70 relative to those detected at day 0 (Fig. 2g). Functional profiles observed in response to Ag85B and TB10.4 in H4:IC31 recipients were very similar and dominated by Th1-cytokine-expressing cells,

with no single IL-22 or IL-17 producing CD4 T cells. However, some participants had detectable IL-2+TNF+IL-22+ or IFNγ+IL-2+TNF+IL-22+ responses (Fig. 2g).

Immunogenicity of H4:IC31 was also confirmed by MIMOSA2, showing that virtually all participants in the H4:IC31 arm and less than 20% of participants in the placebo arm had significantly increased Th1 responses to Ag85B and TB10.4 over baseline (Fig. 2h). Antigen-specific CD8 T cell responses to all antigens tested were low and not significantly modulated by H4:IC31 vaccination (Supplementary Fig. 5f).

Taken together, we showed that H4:IC31 was immunogenic and induced mostly a Th1 CD4 T cell response, although lower frequencies of other cell subsets, such as Th17 and NKT_{like} cells, were also induced.

**Fig. 2 H4:IC31-induced immune responses.** tSNE analysis of total cytokine-expressing lymphocytes observed 70 days post-vaccination in H4:IC31 recipients ($n = 26$) after in vitro restimulation of whole blood with Ag85B (**a**) or TB10.4 (**b**). Mean proportions of each lymphocyte subset among the total cytokine-expressing cells are indicated by the horizontal bars. Error bars represent the 95% CI and colors correspond to those in the tSNE plot. Three cell populations were not identifiable with the flow cytometry antibody panel. **c** Frequencies of CD4 T cells expressing any combination of IFNγ, IL-2, TNF, IL-22, and/or IL-17 for each individual at day 0 (circles) and day 70 (diamonds) randomized to placebo (blue) or H4:IC31 (red). Values above the horizontal floating lines represent $p$ values obtained by comparing responses between day 0 and day 70, calculated by Wilcoxon signed-rank test. **d** Relative proportions of total cytokine-expressing Ag85B- (blue) or TB10.4-specific (orange) CD4 T cells expressing Th1 cytokines (IFNγ, IL-2 and/or TNF), IL-22 or IL-17 measured at day 70 in H4:IC31 recipients. All participants had a detectable response (Fishers' exact test, see "Methods"). **e**, **f** COMPASS polyfunctionality scores for CD4 T cells, stratified by vaccine arm in response to Ag85B (**e**) and TB10.4 (**f**). Changes between day 0 (circles) and day 70 (diamonds) were calculated by Wilcoxon signed-rank test. **g** Heatmap of COMPASS posterior probabilities for detecting an antigen-specific CD4 T cell response on day 70 relative to day 0 for the indicated cytokine-co-expression subsets in H4:IC31 or placebo recipients. Columns correspond to the different cell subsets (shown are the 13 of 24 subsets with detectable antigen-specific response in at least one participant regardless of antigen specificity), identified below the heatmap and color-coded by the cytokines they express (white = none, shaded = present) and ordered by degree of functionality from one function on the left to five functions on the right. Each row in the heatmap represents one participant. **h** Percentage of participants with significant vaccine-induced responses to Ag85B (blue) or TB10.4 (red), calculated by MIMOSA2 (see "Methods" section), based on CD4 T cells expressing any combination of IFNγ, IL-2, TNF, IL-22, and/or IL-17 (left), polyfunctional IFNγ+IL-2+ TNF+ (center) or Th22 (right) cytokines on day 70 relative to day 0.

**In vitro BCG restimulation activates cytokine expression from a broad range of immune cells.** BCG is known to be a potent inducer of Th1 responses but also triggers a broad range of antigen-specific and unspecific responses[8–11]. We applied tSNE at day 70 to all study participants to identify cell populations that expressed IFNγ, IL-2, TNF, IL-22, or IL-17 in response to in vitro BCG restimulation (Fig. 3a and Supplementary Fig. 6a).

Interestingly, conventional CD4 T cells only contributed 33.8% of the total cytokine-expressing response, which was primarily comprised of Th1 and Th22 cells (Th1—18.3%, 95% CI 17.4–19.3; Th22—13.7%, 95% CI 12.7–14.8; other CD4 subsets—1.7%, 95% CI 1.5–1.8; Fig. 3a). Most cytokine-producing CD4 T cells expressed CD26, with some co-expressing CD161 (Supplementary Fig. 6a). The remaining ~75% of BCG-responsive cytokine-producing cells clustered into clear phenotypic cell subsets: 20% were NK cells (5.2%, 95% CI 4.6–5.8 CD56hi and 14.8%, 95% CI 13.3–16.2 CD56int NK cells), 19% (95% CI 17.1–20.8) were γδ T cells (also expressing high levels of CD26 and CD161), and 16.3% (95% CI 14.9–17.7) were phenotypic MAIT cells (T cells co-expressing high levels of CD161 and CD26 (refs. [42–44]) (Fig. 3a). These cell subsets mostly produced IFNγ alone (Supplementary Fig. 6a). Other minor cytokine-producing cell subsets included a mixture of NK and NKTlike cells expressing both IFNγ and TNF (2.9%, 95% CI 2.6–3.3) but also low frequencies of conventional CD8 T cells (1%, 95% CI 0.9–1.1). We also identified one cluster of cells (4.1%, 95% CI 3.6–4.6) that we were not able to define on the basis of marker expression. This subset was negative for all phenotypic markers and mostly produced TNF alone (Fig. 3a and Supplementary Fig. 6b). All participants responded to in vitro BCG restimulation, with no obvious difference in cell subset distribution between study arms (Fig. 3b).

Cluster identity, confirmed by manual gating (Supplementary Figs. 1 and 6b, c), showed that unbiased identification of cell clusters by tSNE was accurate and revealed unknown subsets of cells that would have been overlooked by manual gating.

Overall, these unbiased analyses showed that conventional CD4 T cells only contribute ~30% of cytokine-producing lymphocytes in response to in vitro BCG restimulation.

**BCG revaccination boosts Th1 and Th22 CD4 T cells.** To identify which cell populations were significantly induced by in vivo BCG revaccination, we applied CITRUS to compare responses at day 0 versus day 70 in BCG recipients. Two subsets of cells were significantly induced by BCG revaccination (Fig. 3c), identified as Th1 and Th22 cells by expression analysis of cluster-

defining markers (Supplementary Fig. 7a). Manual gating confirmed these results (Supplementary Fig. 1), showing a significant increase in frequencies of conventional CD4 T cells expressing Th1 cytokines or IL-22 after BCG revaccination (Fig. 3d).

None of the other cell clusters identified by CITRUS were significantly boosted by BCG revaccination. This observation was also confirmed by manual gating; no significant induction of DURTs was observed after BCG revaccination and CD56hi NK cells showed only a modest expansion ($p = 0.036$, Supplementary Fig. 8A). COMPASS analysis of NKTlike, γδ T cells and MAIT cells confirmed that cells mostly produced IFNγ alone and few diverse combinations of cells expressing IFNγ, IFNγ and TNF or Th1 cytokines (Supplementary Fig. 8b). Of interest, we identified a subset of MAIT cells producing IL-22 (Supplementary Fig. 8b).

We also investigated whether frequencies of antigen-specific cells were different in placebo versus BCG recipients at day 70. Four clusters of cells were differentially abundant between participants in the two arms (Fig. 3e, FDR < 0.1): clusters A and B were significantly higher in the BCG arm, while clusters C and D were higher in the placebo arm. Phenotypic analysis (Supplementary Fig. 7b) showed that cluster A and B were Th1- and IL-22-producing CD4 T cells, respectively, which we confirmed by manual gating (Fig. 3f).

The two clusters that were significantly reduced in the BCG compared to the placebo arm did not express markers that allowed verification by manual gating. Cluster C expressed low levels of CD56, IFNγ, and TNF. Cluster D was negative for all phenotypic markers and produced TNF, as observed for the "unknown" cluster observed by tSNE (Fig. 3a).

Since conventional CD4 T cells were the main cell type induced by BCG revaccination, we further analyzed this subset by manual gating. Pre-vaccination BCG-specific CD4 T cell responses were relatively high in all study arms, but these cells were significantly boosted only by BCG revaccination (Fig. 4a). BCG-specific responses encompassed approximately equal frequencies of Th1- and IL-22-expressing CD4 T cells (Fig. 4b), with similar cytokine-co-expression profiles across the study arms (Fig. 4b and Supplementary Fig. 8c). COMPASS analysis showed that BCG revaccination induced heterogeneous functional profiles, including Th1 polyfunctional profiles with an additional predominant single IL-22+ cell population (Fig. 4c) and a modest increase in CD4 T cell polyfunctionality (Fig. 4d). Very few cells expressed IL-17 (Supplementary Fig. 2d) and Th17 cells did not seem to cluster with a defined phenotype (Fig. 3b and Supplementary

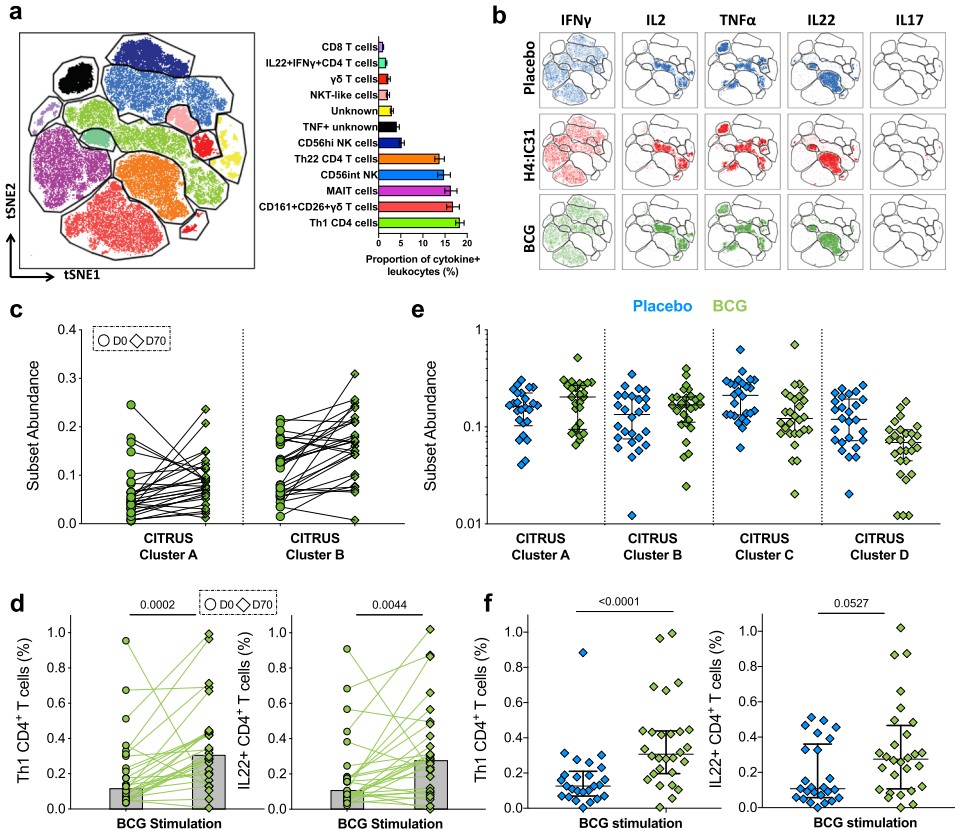

**Fig. 3 BCG-specific immune responses.** tSNE analysis of the total cytokine-expressing lymphocytes observed after BCG restimulation of whole blood collected at day 70 in participants from all study arms (n = 76) (**a**). Mean proportions of each lymphocyte subset among the total cytokine-expressing cells are indicated by the horizontal bars. Error bars represent the 95% CI and colors correspond to those in the tSNE plot. Two cell populations were not identifiable with the flow cytometry antibody panel. **b** The same tSNE plot, disaggregated by cytokine and vaccine arm (placebo, blue, n = 23; H4:IC31, red n = 26; and BCG, green, n = 27). **c** Frequencies of lymphocytes in two BCG-reactive cytokine-expressing cell clusters that were found to be differentially abundant (FDR < 0.1) between day 0 and day 70 in BCG recipients (n = 27) by CITRUS Significance Analysis of Microarrays (SAM) analysis (Supplementary Fig. 7a). **d** Identity of CITRUS clusters shown in c was confirmed by manual gating in FlowJo as BCG-reactive Th1 and IL-22-producing CD4 T cells. Bars denote medians; p values were computed by comparing frequencies between day 0 and day 70 by Wilcoxon signed-rank test. **e** Frequencies of lymphocytes in four BCG-reactive cytokine-expressing cell clusters that were found to be differentially abundant (FDR < 0.1) between placebo (blue) and BCG (green) recipients at day 70 by CITRUS SAM analysis (Supplementary Fig. 7b). **f** Identity of CITRUS cluster A and B shown in e was confirmed by manual gating in FlowJo as BCG-reactive Th1 and IL-22-producing CD4 T cells. Bars denote medians and inter-quartile ranges; p values were computed by comparing frequencies between placebo and BCG recipients by Mann–Whitney U test.

Fig. 6a), nor were they induced by BCG revaccination (Supplementary Fig. 8d).

Despite the high frequencies of BCG-specific CD4 T cell responses detected at baseline, MIMOSA2 analysis confirmed that ~80% of participants who received BCG had significant increases in frequencies of total cytokine-expressing (Fig. 4e), polyfunctional Th1-cytokine-expressing (Fig. 4f), or IL-22-expressing (Fig. 4g) CD4 T cells. Overall, evaluation of vaccine responsiveness was more specific (i.e. low responder rates in placebo arm) when triple positive, IL-2+TNF+IFNγ+CD4 T cells from either H4:IC31 (low baseline responses, Fig. 2) or BCG (high baseline responses, Fig. 4) were analyzed (Supplementary Fig. 9).

Taken together, our analyses showed that BCG revaccination modulated frequencies of Th1 and Th22 CD4 T cells.

## Discussion

We recently showed in the C-040-404 randomized controlled trial that BCG revaccination of QFT-negative adolescents can reduce sustained QFT conversion and that the signal observed in H4: IC31 vaccinees was also suggestive of a possible biological effect[2]. Here, we characterized immune responses induced by these

vaccines in an immunogenicity sub-cohort of C-040-404 participants.

H4:IC31 induced marked increases in polyfunctional CD4 Th1 responses to Ag85B and TB10.4, while BCG revaccination significantly boosted pre-existing CD4 T cells expressing combinations of Th1 cytokines as well as IL-22 alone. Our results that H4: IC31 and BCG vaccination of adolescents induced dominant Th1 responses are consistent with previous clinical trials conducted in adults or infants at the same clinical site in South Africa[6,45,46] and elsewhere[7,47].

However, we also observed that additional functions and cell subsets contributed to these vaccine responses. tSNE analysis showed that CD56+CD3+NKT$_{like}$ cells producing Th1 cytokines contributed ~10% of the total response to Ag85B and TB10.4 and were significantly induced by H4:IC31 70 days after vaccination. NKT cells were the first CD1d-restricted T cells to be described and were shown to co-express NK and T cell-specific markers[48]. NKT cell activation early after *M. bovis* BCG infection of mice has been reported[49]. NKT activation can be mediated via adaptive and innate pathways, such as CD1d-mediated antigen recognition via the TCR as well as response to the cytokines IL-12 and IL-18[49,50]. We stimulated whole blood with Ag85B and TB10.4

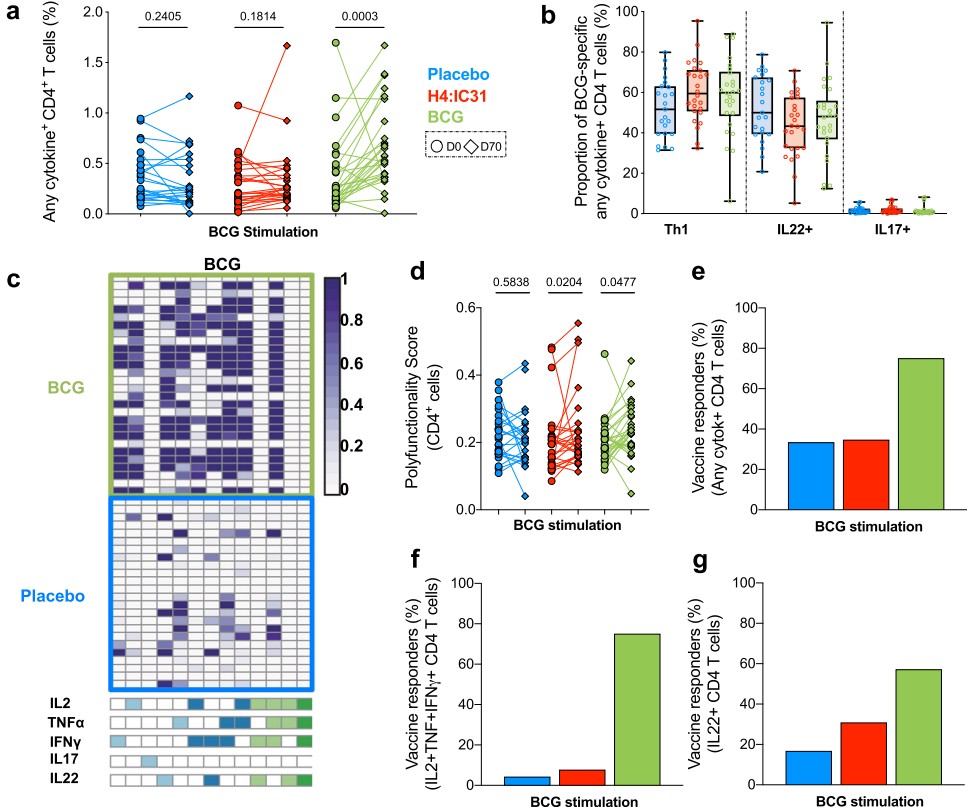

**Fig. 4 Functionality of BCG-specific CD4 T cell immune responses. a** Frequencies of CD4 T cells expressing any combination of IFNγ, IL-2, TNF, IL-22, and/or IL-17 after BCG stimulation at day 0 (circles) and day 70 (diamonds) in the placebo (blue, $n = 24$) H4:IC31 (red, $n = 26$) or BCG (green, $n = 28$) arms. P values were computed by comparing response frequencies between day 0 and day 70 by Wilcoxon signed-rank test. **b** Relative proportions of total cytokine-expressing CD4 T cells expressing either Th1 cytokines (IFNγ, IL-2, and/or TNF), IL-22, or IL-17 after BCG stimulation, measured at day 70 in the placebo (blue, $n = 23$), H4:IC31 (red, $n = 26$), or BCG (green, $n = 27$) arms. Two participants with undetectable response (Fishers' exact test, see "Methods") are not shown. **c** Heatmap of COMPASS posterior probabilities for detecting an antigen-specific CD4 T cell response on day 70 relative to day 0 for the indicated cytokine-co-expression subsets in BCG or placebo recipients. Columns correspond to the different cell subsets identified below the heatmap (shown are the 13 of 24 subsets with detectable antigen-specific response in at least one participant regardless of antigen specificity), color-coded by the cytokines they express (white = none, shaded = present) and ordered by degree of functionality from one function on the left to five functions on the right. Each row in the heatmap represents one participant. **d** COMPASS polyfunctionality score for BCG-stimulated CD4 T cells stratified by vaccine arm in response to BCG (placebo in blue, H4:IC31 in red, and BCG in green). P values represent comparisons between day 0 (circles) and day 70 (diamonds), calculated by Wilcoxon signed-rank test. Each line represents one participant. **e-g** Percentage of participants with significant vaccine-induced responses to BCG, calculated by MIMOSA2 (see "Methods" section), based on any combination of IFNγ, IL-2, TNF, IL-22, and/or IL-17 (**e**), polyfunctional IFNγ+IL-2+TNF+ (**f**), or IL-22-expressing (**g**) CD4 T cells stratified by vaccine arm (placebo $n = 24$, H4:IC31 $n = 26$, BCG $n = 28$).

peptide pools, which to our knowledge cannot be presented by CD1 molecules and thus are unlikely to directly trigger TCR activation on NKT cells[51,52]. Similarly, by-stander activation through cytokines[53], such as dendritic cell-derived IL-12[54], is also unlikely since innate cells cannot respond to naked peptides. In our assay, it is therefore more plausible that activation of Ag85B and TB10.4-specific T cells may lead to innate cell activation, which in turn could mediate cytokine activation of innate T cells, such as NKT cells.

In addition to the well-described Th1 responses, BCG revaccination significantly boosted frequencies of Th22 T cells, which constituted approximately half of the BCG-specific CD4 T cell responses. Interest in a role for IL-22 in mycobacterial containment has increased in recent years (reviewed in ref. [55]). For example, it was shown that IL-22 can restrict mycobacterial growth in macrophages[56,57] and recombinant IL-22 promotes phagolysosome fusion, leading to reduced bacterial burdens in macrophages[58]. IL-22 was also shown to mediate protection against Mtb during the chronic stages of experimental infection of mice with the W-Beijing strain of Mtb, HN878[59]. In non-human primates, IL-22 mRNA expression was reduced in peripheral

blood of Mtb-infected monkeys but increased in lung lymphocytes, bronchial lymph nodes, and spleen, suggesting a role for IL-22 in lymphoid and lung tissues[60]. Finally, several studies have reported IL-22 expression after BCG stimulation of whole blood from individuals living in settings endemic for TB[46,61–63]. Interestingly, IL-22-expressing cells are not readily detected after stimulation of PBMC or in response to peptides[61]. This technical observation has important implications for studies that seek to determine if IL-22 is a correlate of protection, since such studies are typically done on cryopreserved PBMC.

In contrast to our finding that Th17 responses were not modulated by BCG revaccination, a recent study from Rakshit et al.[47] showed increases in Th17 CD4 T cells after BCG revaccination in an Indian cohort. While the magnitude of BCG-reactive Th17 CD4 T cells observed in these two studies was similar, it is not clear why this difference in Th17 responses was observed.

Presence of heterogeneous immune responses pre-vaccination poses challenges in defining vaccine immunogenicity, which cannot be assessed with statistical methods developed to measure vaccine responsiveness in immunologically naïve populations[64].

In contrast to very low baseline responses to Ag85B and TB10.4, immune responses to BCG were already present in all participants at baseline. This was not surprising since all participants received BCG vaccination at birth and stimulation with the whole-cell vaccine may also trigger cross-reactive responses to non-tuberculous mycobacteria. We therefore applied the novel statistical approach, MIMOSA2, to determine responder status by comparing day 70 responses to pre-vaccination responses. When applied to IL-2+TNF+IFNγ+CD4 T cells this method appeared specific and robust to measure vaccine-induced responses to H4:IC31 and BCG, which represented opposites in the spectrum of baseline response magnitudes. Our results support wider application of MIMOSA2 to evaluate immunogenicity of vaccine candidates in populations with pre-existing antigen-specific T cell responses before vaccination.

Overall, "conventional" CD4 T cells contributed about a third of the cytokine-expressing lymphocyte response to in vitro BCG stimulation of whole blood, followed by NK cells, γδ T cells and phenotypic MAIT cells. BCG revaccination was associated with a modest increase in BCG-reactive IFNγ-producing NK cells. We previously showed that neonatal BCG vaccination as well as BCG revaccination of Mtb-infected adults was associated with increased frequencies of BCG-reactive IFNγ-producing NK cells[46]. This result could be explained by NK by-stander activation through cytokines like IL-2, IL-12, and IL-18[46] and/or BCG-mediated epigenetic re-programming of innate cells (mostly monocytes and NK cells), termed "trained immunity"[65,66]. Trained immunity has been proposed as one of the mechanisms associated with non-specific protective effects of BCG[65], which is also consistent with the lower rates of upper respiratory tract infections observed in the BCG arm, compared to placebo and H4:IC31 arms of the C-040-404 trial[67]. We also recently showed that circulating NK cells are increased in abundance and mediate enhanced cytotoxic responses in latent Mtb infection, while a corresponding decrease during active disease was observed[68]. In addition, IFNγ production by NK cells was shown to play an important role in activating and enhancing innate and adaptive immune responses at early stages of pulmonary non-tuberculous mycobacteria infection[69].

It was notable that many other cell subsets typically considered "innate" expressed cytokines in response to in vitro BCG stimulation, including phenotypically defined MAIT cells, γδ T cells, and NKT_like cells. These cell subsets are generally present at low frequencies in peripheral blood, but a high proportion of these DURTs can be activated through cytokine stimulation in a non-specific fashion. We propose that although these responding cell subsets may not typically possess the features of immunological memory known for T and B cells, they may contribute to the milieu in which immune responses to Mtb take place in vivo and thus could play important roles in directing or modulating immunity against Mtb. This might be very important at mucosal sites, including sites of Mtb infection. In this study we were only able to study vaccine-modulated responses in peripheral blood. As such, we propose that these diverse immune cell populations should be taken into consideration when analyzing immune profiles in response to vaccination, ideally also including analyses of immune cells at mucosal sites.

CITRUS, which was developed to analyze mass cytometry datasets, allowed identification of functional cell subsets that were induced by BCG revaccination in an unbiased fashion. Validation of this analysis pipeline by manual gating, hypothesis-driven statistics, the novel statistical approach, MIMOSA2 and other, more established data analysis algorithms, such as COMPASS[38] and tSNE[36], provided confidence that our results are robust.

Despite the substantial contribution to the in vitro BCG response, frequencies and functions of DURTs were not significantly modulated by BCG revaccination. Since the 12-h WB-ICS assay, which was optimized to measure conventional T cells[70] was used to measure these responses, we cannot rule out that other methods may detect changes in DURTs and other non-T cell subsets that we missed. Nonetheless, the lack of DURT cell modulation is an important result for further immunogenicity studies where deeper analysis of these subsets should be performed. For instance, we only measured IFNγ, IL-2, TNF, IL-22, and IL-17 production and may well have missed other important functional features of the vaccine-induced immune response, such as production of other cytokines, chemokines and cytotoxic potential.

Finally, CITRUS analysis also highlighted significantly lower frequencies of two TNF-expressing cell subsets in BCG compared to placebo recipients. One subset appeared to include a mixture of NK and NKT_like cells that co-expressed IFNγ, while the other was negative for all lineage markers and may comprise monocytes or B cells, which have been shown to respond to BCG restimulation[71]. Confirmation of these subsets as BCG-responsive would require more detailed phenotyping.

Taken together, this study proposes a new framework to analyze vaccine-induced immune responses in clinical trials to provide an unbiased and robust view of the global response induced by vaccination. We propose that analysis platforms such as the one presented here are useful to understand complex vaccine-induced interactions between cell subsets that could inform correlates of risk or protection and should be routinely evaluated in clinical studies of TB vaccines and in other research areas.

BCG has been used to prevent severe childhood TB for almost 100 years and is among the most widely used vaccines globally, administered to more than 80% of neonates and infants worldwide[72]. However, the mechanism of BCG-induced protection against TB remains unknown. The C-040-404 study provides a unique opportunity to define vaccine-mediated immune correlates of protection from sustained Mtb infection, which may also confer protection from TB disease, while informing approaches and insights relevant to design and evaluation of new TB vaccines. Such studies are currently ongoing and the results reported here partially informed the experimental approach to discover immune correlates of protection, which includes measurement of multiple immune cell subsets beyond conventional Th1 cells.

Our immunogenicity results suggest that a broader and less biased view of vaccine-induced responses is needed since individual immune subsets most likely do not operate in isolation, and may require collaboration between adaptive and "innate" immune subsets[73]. We propose that the diverse components of this complex "collaborative immunity" are likely to be modulated by vaccination in unconventional ways that can only be revealed by multidimensional analyses.

## Methods

**Trial design and participants.** As reported in detail[2], C-040-404 was a phase IIb, randomized, three-arm, placebo-controlled, partially blinded clinical trial conducted at two South African sites. We enrolled healthy, HIV-uninfected, QFT-negative, 12- to 17-year-old adolescents who received BCG at birth. Adolescents provided written informed assent and parents/legal guardians written informed consent. The protocol was approved by the South African Health Products Regulatory Authority (SAHPRA), formerly the Medicines Control Council of South Africa (Reference number: 20130826) and the Human Research Ethics Committee of the University of Cape Town (SATVI site reference: 471/2013; Emavundleni site reference: 615/2014). More details about regulatory approvals, consent procedures, and inclusion/exclusion criteria are detailed in Nemes et al.[2]. Briefly, adolescents with previously treated or current TB disease, known household TB contact, substance use, or pregnancy were excluded.

Eligible participants were randomized at a 1:1:1 ratio to receive intramuscular placebo (saline) or H4:IC31 (15 µg H4 polyprotein composed of Ag85B and TB10.4 Mtb proteins, Sanofi Pasteur, in 500 nmol IC31 adjuvant, Valneva, Vienna, Austria) on day 0 and day 56, or intradermal BCG (2–8 × 10^5 CFU, Statens Serum Institut) at day 0. The first 90 enrolled participants, approximately 30 per arm, were

included in the immunogenicity and safety cohort. Additional blood was collected from these participants for safety and immunogenicity assays. Participants who did not receive at least one injection, or who had converted their QFT by day 84 but were not QFT-positive at baseline, were excluded since they may have been Mtb-infected at baseline. Immunology data presented in this paper were generated from 29 March 2016 to 31 May 2016 on samples collected from participants of the C-040-404 trial, enrolled between 1 April 2014 and 25 May 2ICS015, at two sites in South Africa. Supplementary Fig. 2 summarizes demographic characteristics of the immunogenicity cohort.

**PBMC ICS assay**. Blood was collected on study days 0 and 70 and PBMCs were isolated and cryopreserved. PBMCs were shipped to Aeras, Cape Town, South Africa, where a 13-color ICS assay was performed, as previously described[74]. Briefly, PBMCs were thawed, suspended in RPMI containing 10% fetal bovine serum (R10, GemCell 100–500), rested overnight, and counted. PBMC stimulation (approximately $1 \times 10^6$ cells per condition) was only performed on samples with at least 70% viability (10 out of 178 samples, 5.6%, were excluded due to poor viability). Peptide pool and BCG stimulation were different. For peptide pool stimulation, PBMC were incubated for 6–7 h with dimethyl sulfoxide (DMSO, Sigma; negative control), phytohemagglutinin (PHA, Remel; 10 μg/mL; positive control), or peptide pools spanning the lengths of Ag85B and TB10.4 (JPT, 1 μg/peptide/mL; 15 mers with an 11 amino acid overlap, pre-diluted in R10 containing CD107a-Alexa488 and GolgiStop and GolgiPlug, at 1 μL each [BD Biosciences, USA]).

For BCG stimulation, PBMC in R10 medium containing co-stimulatory antibodies CD28 and CD49d (BD Biosciences, USA; 1 μL/well) were incubated with R10 medium (negative control), PHA (positive control), and approximately $3 \times 10^5$ CFU/well BCG (SSI) for 2 h, after which CD107a-Alexa488, GolgiStop, and GolgiPlug were added and the cells were incubated for an additional 6–7 h.

Cells were washed with 1× PBS and stained with Aqua Live/Dead Fixable viability dye (Life Technologies, USA) before staining with fluorochrome-conjugated antibodies to surface markers CCR7, CD4, CD14, CD19, and CD45RO, then fixed and permeabilized (Cytofix/Cytoperm, BD Biosciences, USA) and stained for CD3, CD8, IFNγ, IL-2, TNF, IL-22, IL-17, and CD154 (Supplementary Table 1a). Following staining, cells were washed, fixed, and acquired on a BD LSR II flow cytometer (BD Biosciences, US), set to collect up to 150,000 viable CD3+ target cell events for each stimulation condition. All analyses were performed with FlowJo software (FlowJo LLC, USA).

The gating strategy shown elsewhere[2], performed by an initial experienced operator, was identical within participants except for minor adjustments of individual gates. A second operator performed a quality control check and queries were discussed with the scientific supervisor. Data were exported and locked prior to unblinding and statistical analysis. Inclusion criteria for samples to be included in the analyses were as follows: (a) the difference between CD8 IFNγ+ cells in the negative control and CD8 IFNγ+ cells in the positive control should be ≥1% in at least one of the positive control samples; (b) number of acquired CD4+ and CD8+ events each ≥5,000; (c) CD4+ population ≥10% of CD3+ population; (d) CD8+ population ≥5% of CD3+ population. Based on these criteria, 98.8% of samples were included.

**WB-ICS assay**. Venous blood was collected on day 0 and day 70 in sodium heparin-containing tubes and processed for analysis of antigen-specific T cell responses by WB-ICS assay, as previously described[70]. Briefly, within 75 min of collection, WB was stimulated in the presence of co-stimulatory antibodies (anti-CD28 and anti-CD49d, BD Biosciences, 0.25 μg/mL each) with medium alone (unstimulated, RPMI, Lonza), Ag85B peptide pool (15 mers overlapping by 11 amino acids, JPT, 2 μg/mL/peptide), TB10.4 (15 mers overlapping by 11 amino acids, Aeras, 2 μg/mL/peptide), BCG vaccine ($1.2 \times 10^6$ CFU/mL, Statens Serum Institute), or PHA (phytohemagglutinin as a positive control, 5 μg/mL, Bioweb). Following 7 h of incubation at 37 °C, Brefeldin A (10 μg/mL, Sigma-Aldrich) was added and blood was incubated for an additional 5 h. Thereafter, blood was incubated with EDTA (2 mM, Sigma-Aldrich), red blood cells were lysed, and white blood cells fixed using FACSlysing solution (BD Biosciences) prior to cryopreservation in a solution of RPMI, 40% fetal calf serum (FCS, Hyclone) and 10% DMSO (Sigma-Aldrich).

After thawing, cells were permeabilized for 10 min at room temperature in BD Perm/Wash and stained with a panel of pre-titrated monoclonal antibodies (Supplementary Table 1b) diluted in Brilliant Staining buffer (BD Biosciences) for 45–60 min at 4 °C. Compensation for each experiment was calculated from single-stained antibody capture beads (BD Biosciences) acquired with the samples on a BD Fortessa flow cytometer, equipped with four lasers (405, 488, 544, and 633 nm) and 18 detectors for fluorescent parameters. Data were analyzed using FlowJo V10.2 (FlowJo LLC) and SPICE[75] using the gating strategy illustrated in Supplementary Fig. 1.

The gating strategy, applied by an initial experienced operator, was identical within participants with minor adjustments of gates between individual participants. A second operator performed gating quality control and queries were discussed with the scientific supervisor. Data were exported, locked, and analyzed prior to unblinding. Inclusion criteria for samples to be included in the analyses were as follows: (a) unstimulated control was present and interpretable for each set of samples; (b) frequencies of PHA- or BCG-induced total cytokine-expressing CD4 or CD8 T cells were greater than the median + 3MAD (median absolute deviation) of the total cytokine+ CD4 or CD8 T cells of the unstimulated controls

of the entire cohort; (c) for each sample, the frequency of PHA- or BCG-induced total cytokine+ CD4 or CD8 T cells were greater than the frequency of the same cell population in its respective unstimulated control. Based on these criteria, four sample sets were excluded. Except when reporting on proportions of cytokine-producing cells, all data reported from manual gating were background-subtracted (i.e. the frequencies of cytokine+ cells in the unstimulated sample were subtracted from those in the stimulated sample).

**Statistics and reproducibility**. For all cell subsets analyzed, changes in paired response between day 0 and day 70 were calculated by Wilcoxon signed-rank test. When cell subset abundances were compared between groups (i.e. placebo versus BCG), Mann–Whitney test was used in GraphPad Prism v7.0c. Any cytokine response (IFNγ, IL-2, TNF, IL-22, or IL-17) on total, clean lymphocytes (Supplementary Fig. 1) was gated in FlowJo 10.4.2 and exported as .fcs files. Individual .fcs files for Ag85B and TB10.4 stimulation from H4:IC31 arm and BCG stimulation for the three arms were exported with their internal compensation and bi-exponential transformation calculated in the main analysis workspaces.

*tSNE*. Unsupervised nonlinear dimensionality reduction tSNE analysis was performed in FlowJo 10.6.1 on concatenated data from all BCG-stimulated samples from the BCG, H4:IC31, and placebo arms at day 70 and on concatenated data from all Ag85B or TB10.4 stimulated samples from H4:IC31 vaccinees. All fluorescent parameters were selected to cluster populations with 2000 iterations and a perplexity of 40. We used an approximate KNN algorithm (random projection forest, ANNOY) and an FFT interpolation gradient algorithm[76].

*CITRUS*. Significance Analysis of Microarrays (SAM) analysis, a correlative model, was performed in Cytobank with equal sampling of 660 events per file with a minimum cluster size of 5%, a false discovery rate <0.1, and using all markers of the panel.

*COMPASS analysis*. We used the COMPASS1 R package[38] to identify antigen-specific CD4 T cell subsets producing any combinations of IL-2, TNFα, IFNγ, IL-17, and IL-22. The COMPASS model was fit using 3000 iterations and 8 replicates to compute posterior probabilities of antigen-specific cell subsets responses relative to each respective unstimulated control. We also used COMPASS to compute functionality and polyfunctionality scores, which are the mean posterior probability across all cell subsets and the mean posterior probability weighted by the level of functionality of each cell subset, respectively. To estimate vaccine-mediated induction of antigen-specific cells producing different combinations of cytokines above the pre-vaccination response, we used COMPASS to calculate the posterior probabilities of day 70 relative to day 0 (instead of unstimulated).

*Antigen-specific and vaccine responsiveness*. Individual participants were considered to respond to antigen-specific stimulation if the ratio of cytokine positive cells and cytokine negative cells in the stimulated (e.g. BCG, TB10.4, or Ag85B) condition was significantly higher ($p < 0.00001$ by Fisher's exact test[77]) compared to the unstimulated control. Individuals classified as non-responders were excluded from Figs. 2d, 4b and Supplementary Figs. 4c and 8c).

Individual participant's responsiveness to vaccination was evaluated by MIMOSA2. This novel algorithm is an extension of the MIMOSA model[78] and has been extended to include the two-component mixture to model responses at baseline, post-vaccination, and enable testing for the difference in magnitudes of response between baseline and post-vaccination. An individual model was run for each of the three stimulations (e.g. BCG, TB10.4, or Ag85B) and three cytokine subset combinations: (1) CD4 T cells producing any combination of IL-2, TNF, IFNγ, IL-17, or IL-22; (2) IL-2+ TNF+IFNγ+ (triple positive) CD4 T cells; (3) IL-22+CD4 T cells. Response calls were made using a 0.1% FDR threshold.

**Reporting summary**. Further information on research design is available in the Nature Research Reporting Summary linked to this article.

## Data availability

Immunogenicity data from the C-040-404 clinical trial are collated in .csv format on Figshare[79] (https://doi.org/10.25375/uct.12472739.v1). Other relevant data not provided in the published article or supplementary files are available from the corresponding authors upon reasonable request.

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

## Acknowledgements

We thank all participants of this study and their families, and SATVI clinical and laboratory teams. The C-040-404 trial was supported by Aeras, Sanofi Pasteur, the Bill and Melinda Gates Foundation, the Government of the Netherlands Directorate-General for International Cooperation, and the United Kingdom Department for International Development. Sanofi Parsteur supplied the H4 antigen for the H4:IC31 vaccine, and Statens Serum Institut and Valneva (formely Intercell) supplied the IC31 adjuvant. E.N. is a Marylou Ingram Scholar supported by the International Society for Advancement of Cytometry, and V.R. was supported by the Swiss National Foundation.

## Author contributions

E.N., A.M.G., S.G., C.D.G., L.-G.B., M.H., and T.J.S. designed C-040-404 trial. H.G., F.R., L.G.B., and M.H. led clinical procedures. A.T., L.M., M.E., N.B., and S.M. performed laboratory procedures. V.R., E.N., M.M., A.T., G.F., W.F., D.A.H., and M.S. performed data analyses. V.R., E.N., and T.J.S. interpreted the results and wrote the manuscript. The C-040-404 Study Team contributed to the conduct of the clinical trial. All authors read, edited, and approved the manuscript.

## Competing interests

MH discloses a clinical trial grant to University of Cape Town, TJS reports grants to University of Cape Town from Aeras, Sanofi Pasteur, the Bill and Melinda Gates Foundation, the Government of the Netherlands Directorate-General for International Cooperation and the United Kingdom Department for International Development. SG and CDG report being employed by and holding shares and stock options in Sanofi Pasteur. The remaining authors declare no competing interests.

## Additional information

## The C-040-404 Study Team

Charmaine Abrahams[1], Marcelene Aderiye[3], Hadn Africa[1], Deidre Albertyn[7], Fadia Alexander[1], Julia Amsterdam[1], Peter Andersen[8], Denis Arendsen[1], Hanlie Bester[7], Elizabeth Beyers[1], Natasja Botes[1], Janelle Botes[1], Samentra Braaf[1], Roger Brooks[5,9], Yolundi Cloete[1], Alessandro Companie[1], Kristin Croucher[7], Ilse Davids[1], Guy de Bruyn[5,9], Bongani Diamond[1], Portia Dlakavu[1], Palesa Dolo[1], Sahlah Dubel[3], Cindy Elbring[1], Ruth D. Ellis[3], Margareth Erasmus[1], Terence Esterhuizen[1], Thomas Evans[3], Christine Fattore[3], Sebastian Gelderbloem[7], Diann Gempies[1], Sandra Goliath[1], Peggy Gomes[5,9], Yolande Gregg[1], Elizabeth Hamilton[1], Willem A. Hanekom[1], Johanna Hector[1], Roxanne Herling[1], Yulandi Herselman[1],

Robert Hopkins[3], Jane Hughes[1], Devin Hunt[3], Henry Issel[1], Helene Janosczyk[5,9], Lungisa Jaxa[1], Carolyn Jones[1], Jateel Kassiem[1], Sophie Keffers[1], Xoliswa Kelepu[1], Alana Keyser[1], Alexia Kieffer[5,9], Ingrid Kromann[4], Sandra Kruger[1], Maureen Lambrick[7], Bernard Landry[3], Phumzile Langata[1], Maria Lempicki[3], Marie-Christine Locas[5,9], Angelique Luabeya[1], Lauren Mactavie[1], Lydia Makunzi[1], Pamela Mangala[1], Clive Maqubela[1], Boitumelo Mosito[1], Angelique Mouton[1], Humphrey Mulenga[1], Mariana Mullins[1], Julia Noble[1], Onke Nombida[1], Dawn O'Dee[3], Amy O'Neil[5,9], Rose Ockhuis[1], Saleha Omarjee[4], Fajwa Opperman[1], Dhaval Patel[5,9], Christel Petersen[1], Abraham Pretorius[1], Debbie Pretorius[1], Michael Raine[3], Rodney Raphela[1], Maigan Ratangee[1], Christian Rauner[5,9], Susan Rossouw[1], Surita Roux[6], Kathryn Tucker Rutkowski[3], Robert Ryall[5,12], Elisma Schoeman[1], Constance Schreuder[1], Steven G. Self[10], Cashwin September[1], Justin Shenje[1], Barbara Shepherd[3], Heather Siefers[3], Eunice Sinandile[1], Danna Skea[9], Marcia Steyn[1], Jin Su[9], Sharon Sutton[3], Anne Swarts[1], Patrick Syntin[5,9], Michele Tameris[1], Petrus Tyambetyu[1], Arrie van der Merwe[7], Elize van der Riet[1], Dorothy van der Vendt[6], Denise van der Westhuizen[1], Anja van der Westhuizen[7], Elma van Rooyen[1], Ashley Veldsman[1], Helen Veltdsman[1], Emerencia Vermeulen[1], Sindile Wiseman Matiwane[1] & Noncedo Xoyana[1]

[7]Aeras Global TB Vaccine Foundation, Cape Town, South Africa. [8]Statens Serum Institut, Artillerivej 5, 2300 Copenhagen, Denmark. [9]Sanofi Pasteur, 1755 Steeles Ave W, North York, Toronto, ON M2R 3T4, Canada. [10]Statistical Center for HIV Research, Vaccine and Infectious Disease Division, Fred Hutchinson Cancer Research Center, Seattle, WA, USA.

