## [Peer Review File · Communications Biology]

Reviewers' comments:

Reviewer #1 (Remarks to the Author):

Guidelines for Referees can be found here:

http://www.nature.com/authors/editorial_policies/peer_review.html

1. Comments for Author

summary of the manuscript

1. Overall impression of the work

Specific comments, with recommendations for addressing each comment

The manuscript entitled "H4:IC31 and BCG induced immune responses in a prevention of M. tuberculosis infection efficacy trial" by Rozot et al, study immunological responses of the first 90 enrolled participants, 30 per arm, of the C-040-404 trial. Previously the results of C-040-404 trial, were published in NEJM by Nemes et al in 2018 in which, authors showed that BCG vaccination reduced the rate of sustained QFT conversion at 4 IU/mL ($P=0.03$), but not vaccination with H4:IC31.

The present manuscript study the range, function and phenotype of innate and adaptive cellular immune responses conferred by H4:IC31 vaccination or by BCG revaccination. Authors observed that additional functions to Th1 response such as cell subsets contributed to BCG revaccination significantly boosted response of Th22 T cells. The study proposes a new framework to analyse vaccine-induced immune responses in clinical trials to provide a view of the global response induced by vaccination. Authors propose that the diverse components of this complex "collaborative immunity" are likely to be modulated by vaccination in unconventional ways that can only be revealed by multidimensional analyses. And authors conclude that where H4:IC31-specific responses would be predominated by Th1 cytokine-expressing CD4 T cells, while BCG would induce a broader range of immune subsets and functions.

Manuscript would be much more attractive if the authors will focus results and discussion on the direct comparing the immunological differences between regimen of vaccination that did prevent Mtb infection, measured as less sustained QFT conversion (BCG revaccination), as compared by the control and H4: C31 vaccine, that did not prevent Mtb infection. After all the exhaustive immunological study of the entire manuscript at the end of the discussion L457, is said, "the mechanism of BCG-induced protection against TB remains unknown" which greatly reduces the reader's expectations, who is looking for an immunological correlation between BCG revaccination and decreased QFT sustained conversion. To include a final discussion on the relevance of the results obtained for future immunological studies, with other live attenuated vaccines currently in clinical trials, could help to increase the relevance the presented results in the field of TB vaccines.

Some Comments

Title: should be focused on BCG revaccination immunological results since to include H4:IC31 that did not prevent Mtb infection, does not add anything.

Discussion, Line 341: should say "reduce " and no "protect" since BCG vaccine reduced the rate of sustained QFT conversion.

That H4:IC31 induced increases in polyfunctional CD4 Th1 responses to Ag85B and TB10.4, should be interpreted as vaccine taken since are the Ags contained in H4: C31 vaccine and not too much discussion is needed. Discussion should be concentrated on the immunological results of BCG revaccination that could be of interest to determine the correlates of prevention of infection results of the Clinical Trial.

Legend of Figure 1 say "in three vaccine arms ," should be placebo and two vaccine arms.

Participants excluded of the study. L485-487 : “,... who had converted their QFT by day 84 but, were not QFT-positive at baseline, were excluded since they may have been Mtb-infected at baseline” were no all the participants QFT-negatives at the beginning of the study? (L473)

Reviewer #2 (Remarks to the Author):

This research is an important contribution to the field, illustrating how multiparamater unsupervised analyses can lead to discovery in the TB vaccine field during clinical trial. The relevance of this can be extended to other vaccine studies. The analyses appear to have been conducted very well. Some comments seeking clarification, or suggestions to improve the writing are listed below:

Abstract

1. It would be helpful to improve the context in the abstract, firstly of the trial and secondly being clear this manuscript presents responses measured post-vaccination and when these responses were measured.
2. L64/65 – explain what H4:IC31 is, and that this and BCG are vaccines.
3. L70 – which cells subsets are “such immune subsets” and L72 “these cells”.
4. It is difficult to follow the conclusion

Introduction

1. L1 Would a vaccine that prevents Mtb infection also by default prevent disease? A vaccine that prevent disease does not necessarily prevent infection, but a vaccine that prevents infection must surely also prevent TB disease. This opening sentence does not flow well to the next sentence.
2. L82 explain that Ag85B and TB10.4 are antigen from Mtb.
3. L86 explain that BCG is live.
4. L94 explain how 4IU/ml differs from the manufacturer’s threshold
5. L95 with neither meeting “significance by” standard ...?
6. L90-99 – have these data been published elsewhere? (I think so, in reference 2). If they have, please include the citation and less detail. If not, they would be better placed in the results rather than introduction and state in the methods how many participants were in each study arm to measure efficacy?
7. L108 – while CD4 T cell deficiency occurs during HIV infection, many other cell populations are depleted; therefore it lends only weak support to your argument.
8. L115-116 simply because CD4 T cells are important for protection in a naive individual, it does not mean that by default a protective memory immune response must be generated from CD4 T cells. There is a lot of evidence in the literature that suggests Th1 cytokines induced by vaccines do not correlate with protection.

Results

1. L144 abbreviations PBMC and ICS
2. L147-152 – different IL-17 and IL-22 antibody clones were used between WB and PBMCs, this should be clearly stated.
3. L163/L178-179 – you should state clearly which lymphocytes you have eliminated from your analysis (eg ILCs, B cells etc) from PBMCs. Why were these ‘dumps’ excluded from the WB analysis? Did you measure the contribution of the excluded populations to cytokine production?

Discussion

1. L348 – can you say the Th1 responses were dominant, as you have restricted your analyses to only Th1/Th17/Th22 responses?

2. It may be useful to comment on the prevalence on the non-conventional populations in blood and whether you necessarily would detect their activation/change in frequencies that may occur in other sites, including the lung.

Supplementary Data

1. In Supplementary Figure 1, where in the gating strategy do you gate our LD/CD19 and CD14 positive cells or was this WB and not PBMCs? It is unclear in L163.

Methods

1. L496 how many samples were excluded for poor viability?
2. L521-525 how many samples were excluded based on these criteria?
3. L553-560 how many samples were excluded based on these criteria?

General comment – this manuscript would be easier to read with better paragraph structure.

Reviewer #1:

The manuscript entitled “H4:IC31 and BCG induced immune responses in a prevention of M. tuberculosis infection efficacy trial” by Rozot et al, study immunological responses of the first 90 enrolled participants, 30 per arm, of the C-040-404 trial. Previously the results of C-040-404 trial, were published in NEJM by Nemes et al in 2018 in which, authors showed that BCG vaccination reduced the rate of sustained QFT conversion at 4 IU/mL ($P=0.03$), but not vaccination with H4:IC31.

Response: We thank the reviewer for this summary and would like to clarify few points regarding the published manuscript (*Nemes.et.al.*), which we think are important to interpret the work under consideration. BCG revaccination was associated with reduced rates of sustained QFT conversion above 0.35 IU/mL and QFT conversion above 4IU/mL (sustained QFT conversion above 4IU/mL was not evaluated), with efficacy of ~45% for both end-points, which was significant at 95% confidence interval (CI). Vaccination with H4:IC31 showed efficacy for the same end-points (30% and 35% efficacy, respectively), but these were only significant at the 80% CI, which was the protocol-defined criterion for efficacy. However, protection conferred by H4:IC31 was not significant at the more rigorous and standard 95% CI. As such, we cannot definitively rule out that H4:IC31 showed some efficacy.

The present manuscript study the range, function and phenotype of innate and adaptive cellular immune responses conferred by H4:IC31 vaccination or by BCG revaccination. Authors observed that additional functions to Th1 response such as cell subsets contributed to BCG revaccination significantly boosted response of Th22 T cells. The study proposes a new framework to analyse vaccine-induced immune responses in clinical trials to provide a view of the global response induced by vaccination. Authors propose that the diverse components of this complex “collaborative immunity” are likely to be modulated by vaccination in unconventional ways that can only be revealed by multidimensional analyses. And authors conclude that where H4:IC31-specific responses would be predominated by Th1 cytokine-expressing CD4 T cells, while BCG would induce a broader range of immune subsets and functions.

Manuscript would be much more attractive if the authors will focus results and discussion on the direct comparing the immunological differences between regimen of vaccination that did prevent Mtb infection, measured as less sustained QFT conversion (BCG revaccination), as compared by the control and H4: C31 vaccine, that did not prevent Mtb infection.

Response: Unfortunately, this trial was not designed (nor powered) for a head-to-head comparison between BCG and H4:IC31, but to compare each vaccine arm to a shared placebo arm. In keeping with the predefined statistical design of the trial, and to protect against biased reporting, we therefore feel that it is important to report in the main figures the results for each vaccine separately. Further, because the vaccines are very different, and because we cannot say definitively that H4:IC31 was not efficacious, we contend that showing immunogenicity results for both regimens is correct and of interest to the reader. However, we agree that a side-by-side comparison of immune responses induced by the two vaccines is of relevance for understanding the results and have now added the head-to head comparison for the 2 vaccine arms and the placebo arm for the responses measured in CD4 T cells, CD8 T cells, gd T cells, MAIT cells and NKT-like cells and NK cells as Supplementary figure 5.

Supplementary figure 5

After all the exhaustive immunological study of the entire manuscript at the end of the discussion L457, is said, “the mechanism of BCG-induced protection against TB remains unknown” which greatly reduces the reader's expectations, who is looking for an immunological correlation between BCG revaccination and decreased QFT sustained conversion. To include a final discussion on the relevance of the results obtained for future immunological studies, with other live attenuated vaccines currently in clinical trials, could help to increase the relevance the presented results in the field of TB vaccines.

Response: We thank the reviewer for this comment. Unfortunately, it is not possible to infer correlates of protection from the results presented in this paper. This is because the immunogenicity analyses in this paper were performed in a subset of the trial participants. As already emphasized in the discussion of the manuscript ‘We propose that analysis platforms such as the one presented here are useful to understand complex vaccine-induced interactions between cell subsets that could inform correlates of risk or protection and should be routinely evaluated in clinical studies of TB vaccines and in other research areas’. We have now added the following statement to clarify how the results reported here are relevant to the current efforts aimed at discovering immune correlates of protection:

“Such studies are currently ongoing and the results reported here partially informed the experimental approach to discover immune correlates of protection, which includes measurement of multiple immune cell subsets beyond conventional Th1 cells”.

Some Comments

Title: should be focused on BCG revaccination immunological results since to include H4:IC31 that did not prevent Mtb infection, does not add anything.

Response: We respectfully disagree with the reviewer because, as stated above, we cannot definitively rule out that H4:IC31 showed some efficacy in the trial. As such it is important that we do not only show “positive” results by omitting the H4:IC31 data, which would bias the results presented in this paper. We believe that results from the H4:IC31 are important even if the efficacy signal was not as strong as for BCG, since they can provide useful information about other sub-unit vaccine candidates formulated with the same adjuvant (e.g. H56:IC31), which are currently being tested in other clinical trials.

Discussion, Line 341: should say “reduce “ and no “protect” since BCG vaccine reduced the rate of sustained QFT conversion.

Response: Thank you, this has been changed in the text.

That H4:IC31 induced increases in polyfunctional CD4 Th1 responses to Ag85B and TB10.4, should be interpreted as vaccine taken since are the Ags contained in H4: C31 vaccine and not too much discussion is needed. Discussion should be concentrated on the immunological results of BCG revaccination that could be of interest to determine the correlates of prevention of infection results of the Clinical Trial.

Response: As mentioned above, we respectfully disagree with the reviewer because we cannot definitively rule out that H4:IC31 showed some efficacy in the trial. In this manuscript we are reporting the immunogenicity results from the C-040-040 trial and, as mentioned previously, cannot infer which aspects will correlate with vaccine protection.

Legend of Figure 1 say “in three vaccine arms ,” should be placebo and two vaccine arms.

Response: Thank you for pointing this out. It has been modified.

Participants excluded of the study. l485-487 : “ ,.... who had converted their QFT by day 84 but, were not QFT-positive at baseline, were excluded since they may have been Mtb-infected at baseline” were not all the participants QFT-negatives at the beginning of the study? (L473)

Response: Enrolled participants were QFT negative at screening and study start (D0). However, because it can take up to ~40 days from TB exposure to develop the T cell immune response measured by QFT, it is possible that some individuals were already infected at D0, despite a negative QFT.

To ensure exclusion of individuals who were potentially infected at the time of vaccination, a “wash-out” period was applied to exclude those who converted to a positive QFT by day 84.

Reviewer #2:

This research is an important contribution to the field, illustrating how multiparameter unsupervised analyses can lead to discovery in the TB vaccine field during clinical trial. The relevance of this can be extended to other vaccine studies. The analyses appear to have been conducted very well. Some comments seeking clarification, or suggestions to improve the writing are listed below:

Abstract

1. It would be helpful to improve the context in the abstract, firstly of the trial and secondly being

clear this manuscript presents responses measured post-vaccination and when these responses were measured.

Response: We have carefully revised the abstract to respond to this comment, while ensuring that we do not exceed the limit of 150 words.

2. L64/65 – explain what H4:IC31 is, and that this and BCG are vaccines.

Response: These explanations have been added to the abstract

3. L70 – which cells subsets are “such immune subsets” and L72 “these cells”.

Response: This has been modified to be clearer. It refers to donor-unrestricted T cells.

4. It is difficult to follow the conclusion

Response: This has been modified to the following: “BCG, which demonstrated efficacy against sustained Mycobacterium tuberculosis infection, modulated multiple immune cell subsets, in particular conventional Th1 and Th22 cells, which should be investigated in discovery studies of correlates of protection.”

Introduction

1. L1 Would a vaccine that prevents Mtb infection also by default prevent disease? A vaccine that prevent disease does not necessarily prevent infection, but a vaccine that prevents infection must surely also prevent TB disease. This opening sentence does not flow well to the next sentence.

Response: It is not definitely true that a vaccine that prevents infection must also prevent disease, unless the vaccine is 100% efficacious. For a partially protective vaccine, such as BCG, the individuals that are protected against infection may be the “special” ones that would not have progressed to TB disease anyway. As such, we cannot assume that protection against infection equals protection against disease. We have now modified the sentence by removing the mention of TB disease to make it less confusing: “A vaccine that can prevent infection with Mycobacterium tuberculosis (Mtb) could have a major impact on the tuberculosis (TB) epidemic”

2. L82 explain that Ag85B and TB10.4 are antigen from Mtb.

Response: We have modified the sentence as follows: “The H4:IC31 vaccine candidate comprises a fusion protein of mycobacterial antigens Ag85B and TB10.4 formulated in the IC31 adjuvant”

3. L86 explain that BCG is live.

Response: This has been added.

4. 94 explain how 4IU/ml differs from the manufacturer’s threshold

Response: We have now modified the text to make this clearer: “Neither vaccine protected against the primary C-040-404 trial endpoint, initial infection with Mtb, defined as conversion to a positive QFT at the manufacturers’ threshold (≥ 0.35 IU/mL). Efficacy of H4:IC31 against sustained QFT conversion, defined as QFT conversion without reversion to a negative QFT for three consecutive tests, was 30.5% (95%CI -15.8 to 59.3, $p=0.16$), and efficacy against conversion at the exploratory cut-off of 4IU/mL was 34.5% (95%CI -12.1 to 62.3, $p = 0.13$).”

5. L95 with neither meeting “significance by” standard ...?

Response: This has been rephrased: “Neither of these efficacy estimates met the standard 95% CI criteria for statistical significance.”

6. L90-99 – have these data been published elsewhere? (I think so, in reference 2). If they have, please include the citation and less detail. If not, they would be better placed in the results rather than introduction and state in the methods how many participants were in each study arm to measure efficacy?

Response: We thank the reviewer for this comment. Even though these results have been published (reference 2, now added), we believe it is important to retain the summary of the main trial findings in the introduction, since this is critical to set the context and to interpret the results reported in this manuscript (without expecting the reader to be familiar with previous work or reading the published work first).

7. L108 – while CD4 T cell deficiency occurs during HIV infection, many other cell populations are depleted; therefore it lends only weak support to your argument.

Response: We agree that HIV infection alters cell populations other than CD4 T cells. To make this statement more definitive, we have now added the additional statement: “possibly because HIV has been shown to preferentially impair M.tb-specific T cells^{14-16”}. This refers to three studies that have shown that Mtb-specific CD4 T cells are depleted to a greater degree than CD4 T cells specific to other infectious organisms.

- Geldmacher C, Ngwenyama N, Schuetz A, Petrovas C, Reither K, Heeregrave EJ, Casazza JP, Ambrozak DR, Louder M, Ampofo W, Pollakis G, Hill B, Sanga E, Saathoff E, Maboko L, Roederer M, Paxton WA, Hoelscher M, Koup RA. Preferential infection and depletion of Mycobacterium tuberculosis-specific CD4 T cells after HIV-1 infection. *J Exp Med*. 2010 Dec 20;207(13):2869-81. doi: 10.1084/jem.20100090. Epub 2010 Nov 29. PMID: 21115690
- Christof Geldmacher , Alexandra Schuetz, Njabulo Ngwenyama, Joseph P Casazza, Erica Sanga, Elmar Saathoff, Catharina Boehme, Steffen Geis, Leonard Maboko, Mahavir Singh, Fred Minja, Andreas Meyerhans, Richard A Koup, Michael Hoelscher. Early Depletion of Mycobacterium Tuberculosis-Specific T Helper 1 Cell Responses After HIV-1 Infection. *J Infect Dis*. 2008 Dec 1;198(11):1590-8. doi: 10.1086/593017.
- Amelio, P. et al. HIV Infection Functionally Impairs Mycobacterium tuberculosis-Specific CD4 and CD8 T-Cell Responses. *J Virol* 93, doi:10.1128/JVI.01728-18 (2019).

8. L115-116 simply because CD4 T cells are important for protection in a naive individual, it does not mean that by default a protective memory immune response must be generated from CD4 T cells. There is a lot of evidence in the literature that suggests Th1 cytokines induced by vaccines do not correlate with protection.

Response: We agree with the reviewer that this statement is arguable, however here we are summarizing the current thinking in the TB field and what the rationale is behind current vaccine candidate design in light of the absence of correlates of protection. We added further literature support to this point in the text:

- Andersen, P. & Scriba, T. J. Moving tuberculosis vaccines from theory to practice. *Nat Rev Immunol* 19, 550-562, doi:10.1038/s41577-019-0174-z (2019).
- O'Garra, A. et al. The immune response in tuberculosis. *Annu Rev Immunol* 31, 475-527, doi:10.1146/annurev-immunol-032712-095939 (2013).

Results

1. L144 abbreviations PBMC and ICS

Response: This has been added.

2. L147-152 – different IL-17 and IL-22 antibody clones were used between WB and PBMCs, this should be clearly stated.

Response: This has been clarified.

3. L163/L178-179 – you should state clearly which lymphocytes you have eliminated from your analysis (eg ILCs, B cells etc) from PBMCs. Why were these ‘dumps’ excluded from the WB analysis? Did you measure the contribution of the excluded populations to cytokine production?

Response: We thank the reviewer for this comment. We provided a gating strategy in Supplementary figure 1 showing which cells were excluded (essentially large myeloid cells, doublets and auto-fluorescent cells) from analysis of the WB-ICS assay, which was the main focus of this manuscript. In the WB-ICS panel we did not include a “dump” channel with CD14 and CD19, and therefore these cells may contribute to the cytokine production detected in “total” lymphocytes, notably to the “unknown” populations revealed by the tSNE analysis.

Since the PBMC-ICS panel was designed to detect antigen-specific T cells induced by vaccination, a dump channel was included to reduce non-specific signal from dead cells (which are not an issue in WB-ICS because fresh cells are stimulated immediately after blood draw and are then fixed after stimulation), B cells and monocytes. Results from the PBMC-ICS panel have only been analysed by manual gating, with a focus on the primary immunogenicity endpoint of the trial, namely conventional T cells (reported in Nemes et al), and not directly compared with WB-ICS, except for the data reported in Supplementary Figure 2, which also focus on conventional T cells only.

Discussion

1. L348 – can you say the Th1 responses were dominant, as you have restricted your analyses to only Th1/Th17/Th22 responses?

Response: We thank the reviewer for this comment, which is important. We feel that this statement is in fact reasonable since it is strongly supported by prior evidence. Early work performed in mouse models of H4:IC31 or H1:IC31 vaccination showed that the IC31 adjuvant preferentially directed immune responses to a very robust Th1 response, with virtually no Th2 cytokines detected (*i.e.* EM Ager.et.al., Vaccine, 2006). This was not surprising given that IC31 has been designed as a Th1 priming adjuvant. Several human trials of H4:IC31 or H1:IC31 have also demonstrated the very focused Th1 response, and virtually no Th17 or Th22 responses, induced by these vaccines. In addition, using a set of 4 ELISpot assays we previously showed that M.tb-specific T cells from healthy M.tb-infected South African adolescents are highly polarized towards Th1, with very little, if any IL-5, IL-10 or IL-17 production (Cecilia S. Lindestam Arlehamn et. al. Plos Pathogen, 2016). Furthermore, Amelio P. and colleagues (PLoS Negl Trop Dis, 2017) showed that Th2 responses were almost non-existent in a cohort of South African participants, compared to those observed in a Tanzanian cohort where helminth co-infection is common. The statement that “the Th1 responses were dominant” is therefore not unjustified in our view and would contend that the addition of Th2 markers would only have confirmed undetectable CD4 T cells responses.

2. It may be useful to comment on the prevalence on the non-conventional populations in blood and whether you necessarily would detect their activation/change in frequencies that may occur in other sites, including the lung.

Response: This is an important point. Our work is unfortunately limited to analysis of peripheral blood and we cannot comment on the frequencies of non-conventional populations of cells induced by vaccination in humans at mucosal sites or at the most common site of TB disease, the lung. We have now added some text to the discussion to acknowledge this: “We propose that although these responding cell subsets may not typically possess the features of immunological memory known for T and B cells, they may contribute to the milieu in which immune responses to Mtb take place in vivo and thus could play important roles in directing or modulating immunity against Mtb. This might be very important at mucosal sites, including sites of Mtb infection. In this study we were only able to study vaccine-modulated responses in peripheral blood. As such, we propose that these diverse immune cell populations should be taken into consideration when analysing immune profiles in response to vaccination, ideally also including analyses of immune cells at mucosal sites.” We note that we are currently performing a follow-up study of BCG revaccination which includes samples collected by Broncho-alveolar lavage, to assess the immunity conferred by the BCG revaccination at the site of infection.

Supplementary Data

1. In Supplementary Figure 1, where in the gating strategy do you gate our LD/CD19 and CD14 positive cells or was this WB and not PBMCs? It is unclear in L163.

Response: This figure refers to the whole blood ICS assay, which did not include the LD/CD19 and CD14 positive cell gates. This has now been clarified in the Supplementary Figure 1 legend.

Methods

1. L496 how many samples were excluded for poor viability?

Response: 10 of 178 PBMC samples were excluded for poor viability for the study (5.6%). This information has been added in the corresponding methods section.

2. L521-525 how many samples were excluded based on these criteria?

Response: 2 samples were not included based on the inclusion criteria (98.8% were included). This has been added in the relevant section.

3. L553-560 how many samples were excluded based on these criteria?

Response: Based on these criteria, 4 samples were excluded. This has also been added in the method section.

General comment – this manuscript would be easier to read with better paragraph structure.

Response: We thank the reviewer for the suggestion and have tried to improve the paragraph structure across the manuscript.

REVIEWERS' COMMENTS:

Reviewer #1 (Remarks to the Author):

The authors have clarified my doubts and answered my questions.